# Procymidone Application Contributes to Multidrug Resistance of *Botrytis cinerea*

**DOI:** 10.3390/jof10040261

**Published:** 2024-03-29

**Authors:** Zhaochen Wu, Chuxian Yu, Qiuyan Bi, Junting Zhang, Jianjun Hao, Pengfei Liu, Xili Liu

**Affiliations:** 1Department of Plant Pathology, China Agricultural University, Beijing 100193, China; zhaochenwu@cau.edu.cn (Z.W.); s20233193296@cau.edu.cn (C.Y.); 18843008100@163.com (J.Z.); seedling@cau.edu.cn (X.L.); 2Institute of Plant Protection, Hebei Academy of Agricultural and Forestry Sciences, Ministry of Agriculture, Baoding 071000, China; 0304biqiuyan@haafs.org; 3School of Food and Agriculture, University of Maine, Orono, ME 04469, USA; jianjun.hao1@maine.edu

**Keywords:** transcription, multidrug resistance, single-point mutations, molecular docking

## Abstract

The necrotrophic pathogen *Botrytis cinerea* infects a broad range of plant hosts and causes substantial economic losses to many crops. Although resistance to procymidone has been observed in the field, it remains uncertain why procymidone is usually involved in multidrug resistance (MDR) together with other fungicides. Nine mutants derived from the *B. cinerea* strain B05.10 through procymidone domestication exhibited high resistance factors (RFs) against both procymidone and fludioxonil. However, the fitness of the mutants was reduced compared to their parental strain, showing non-sporulation and moderate virulence. Furthermore, the RFs of these mutants to other fungicides, such as azoxystrobin, fluazinam, difenoconazole, and pyrimethanil, ranged from 10 to 151, indicating the occurrence of MDR. Transcriptive expression analysis using the quantitative polymerase chain reaction (qPCR) revealed that the mutants overexpressed ABC transporter genes, ranging from 2 to 93.7-fold. These mutants carried single-point mutations W647X, R96X, and Q751X within BcBos1 by DNA sequencing. These alterations in BcBos1 conferred resistance to procymidone and other fungicides in the mutants. Molecular docking analysis suggested distinct interactions between procymidone and Bos1 in the *B. cinerea* standard strain B05.10 or the resistant mutants, suggesting a higher affinity of the former towards binding with the fungicide. This study provides a comprehensive understanding of the biological characteristics of the resistant mutants and conducts an initial investigation into its fungicide resistance traits, providing a reference for understanding the causes of multidrug resistance of *B. cinerea* in the field.

## 1. Introduction

The necrotrophic pathogen *Botrytis cinerea* is the causal agent of gray mold, which infects over 1000 plant species [1]. It employs a variety of pathogenicity factors such as cell wall-degrading enzymes, phytotoxins (botrydial and botcinic acid), phytohormones, reactive oxygen species, oxalic acid, and small RNAs, which contribute to the infection process and decay of plant tissues [2].

Chemical control using fungicides has been the most rapid and effective method for preventing and controlling gray mold disease since the 1960s. However, gray mold frequently develops resistance to fungicides with various modes of action in the field. The pathogen responsible, *B. cinerea*, exhibits significant genetic variability, posing a high risk of resistance (Fungicide Resistance Action Committee, FRAC, https://www.frac.info/, since 2012).

Resistance to fungicides commonly occurs in the field with *B. cinerea*, which is not only toward single fungicides or a specific type of fungicide based on their mode of action (MOA), but also toward various fungicides with different MOAs [3]. In 1983, shortly after the use of benzimidazole fungicides and dicarboximides fungicides, multidrug-resistant (MDR) strains of *B. cinerea* that were simultaneously resistant to two types of fungicides appeared [4].

In recent years, MDR in *B. cinerea* has been frequently reported. Current knowledge of the MDR mechanisms includes mutations at multiple target sites due to the use of various fungicides, overexpression of metabolic enzyme genes due to exposure to uncouplers, and overexpression of efflux protein genes. The latter is more frequently associated with MDR—the fungicide efflux transporters include ABC and MFS transporters in *B. cinerea* [5]. *Botrytis cinerea* contains 14 ABC efflux proteins (BcatrA-BcatrN) and two MFS efflux proteins (Bcmfs1, BcmfsM2). Among them, four ABC proteins are related to the MDR phenotype, namely BcatrB, BcatrD, BcatrK, and BcatrO [6,7,8]. BcatrB, BcatrD, BcatrK, Bcmfs1, BcmfsM2, and BcmfsG have been reported to be involved in the efflux of fungicides and phytoalexin [9,10]. 

The earliest report on MDR can be traced back to the 1980s [11]. Shortly after the use of benzimidazole and dicarboximide fungicides, MDR strains of *B. cinerea* emerged that simultaneously exhibited resistance to both types of agents [12,13]. Meanwhile, fludioxonil resistance has been frequently reported to be present in the resistant populations in the field [13]. It is unclear if the fludioxonil application is linked to the emergence of MDR. Nonetheless, the actual causes of MDR in the field remain unclear and need to be further studied.

Procymidone (Pro), as a dicarboximides (DCF) fungicide, inhibits mycelial growth and conidial germination by disrupting osmoregulation [12], which is frequently associated with resistant populations of *B. cinerea*. ABC transporters in *B. cinerea* are upregulated when exposed to fungicides such as fludioxonil and procymidone [13]. Procymidone, fludioxonil, and some other chemicals have been reported to induce fungicide resistance through the overexpression of ABC transporters in *B. cinerea*, but there is no direct evidence that these agents lead to MDR. This study intended to elucidate the possibility of procymidone inducing MDR in *B. cinerea* and, if proven, to analyze the resistance mechanisms.

## 2. Materials and Methods

### 2.1. Fungicides

The fungicides used included azoxystrobin (98.0%, Syngenta Biotechnology, Shanghai, China), salicylhydroxamic acid (SHAM) (99%, Shanghai Marklin Biochemistry, Shanghai, China), boscalid (99%, BASF Corporation, Nanjing, China), fluazinam (98.4%, Ishihara Sangyo Kaisha Ltd., Osaka, Japan), fludioxonil (97.9%, Syngenta Biotechnology, Co., Ltd., Shanghai, China), difenoconazole (98.4%, Yulong Chemical Industrial Co., Ltd., Hangzhou, China), pyrimethanil (93%, Jiangsu Limin Chemical Industrial Co., Ltd., Xinyi, China), and procymidone (98.5% active ingredient, Jiangxi Heyi Chemicals Company Ltd., Jiujiang, China), and were dissolved in dimethyl sulfoxide (DMSO) to prepare a stock solution (1 × 10^5^ µg/mL) and kept in darkness at −20 °C.

### 2.2. In Vitro Selection of Procymidone-Resistant B. cinerea

*Botrytis cinerea* strains B05.10 (obtained from Germany, Institut für Botanik, Westfälische Wilhelms-Universität, Münster) and 223 (collected from Yinlong Soil Farm in Shanghai in April 2016) were used as a standard resistant strain and sensitive strain, respectively. These two strains were both sensitive to procymidone. Procymidone-resistant mutants were selected by exposing these two isolates to procymidone. *B. cinerea* mycelial plugs (5 mm diameter) were excised from the culture and then transferred onto potato dextrose agar (PDA) plates containing procymidone at 25, 50, and 100 μg/mL. Mutants were selected and transferred to a new PDA plate. In each transfer, the procymidone concentration was gradually increased from 200, 400 to 600 μg/mL, and each concentration was substituted once. The effective concentration for 50% mycelial growth inhibition (EC_50_) was measured [12].

### 2.3. Resistance Level and Stability of Procymidone-Resistant B. cinerea Mutants

The sensitivity of *B. cinerea* was examined by following Song et al.’s method [14] with slight modifications. EC_50_ values of both procymidone-resistant mutants and their wild-type parents were derived as previously described. The resistance factor (RF) was calculated as the EC_50_ of the mutant/EC_50_ of the parental strain. To assess the stability of the resistant mutants, mycelial plugs from the margins of actively growing colonies were subjected to 10 successive transfers on fresh and fungicide-free PDA plates. Mutants were cultured for 5 d at 18 °C in darkness for each transfer, and the EC_50_ values were measured on the 1st and 10th transfers. The stability of resistance was represented by the factor of sensitivity change (FSC), where FSC = the ratio of RF values at the 1st to 10th transfers [15]. Each mutant was evaluated in triplicate, and the experiment was conducted twice.

### 2.4. Cross-Resistance

All resistant mutants and their parental strain B05.10 of *B. cinerea* were tested for sensitivity to fungicides, having different modes of action. The selected fungicides included fludioxonil (5, 10, 20, 50, 100, and 150 µg/mL), azoxystrobin (1, 5, 10, 20, and 50 µg/mL), fluazinam (0.1, 0.5, 1, 5, and 10 µg/mL), boscalid (0.5, 1, 5, 10, and 20 µg/mL), difenoconazole (0.5, 1, 5, 10, 20, and 50 µg/mL), and pyrimethanil (0.1, 0.5, 1, 5, and 10 µg/mL). Each treatment had three replicates (plates), and the experiment was conducted twice.

### 2.5. Mycelial Growth Rate

The activated resistant mutants and their parent strains were cultured in the dark at 18 °C for 3 d. Subsequently, 5 mm mycelial plugs cut from the edge of the colonies were transferred to PDA plates. The cultures were then incubated in the dark at 18 °C. The colony diameter of each strain was measured after 3 d using the cross-intersection method [15]. Each treatment was conducted three times.

### 2.6. Mycelial Growth under Different Temperatures

The procymidone-resistant mutants and corresponding parental isolates were incubated on PDA plates at 4, 14, 18, 25, and 30 °C. Colony diameters were measured in two perpendicular directions after 3 days of incubation in the dark. Each combination of isolate and temperature was represented by four replicated plates.

### 2.7. Conidial Production of B. cinerea

The resistant mutants and their parent strains of *B. cinerea* were cultured in the dark at 18 °C for 3 d. After the incubation, mycelial plugs cut from the edge of the colonies were transferred to a carrot agar (CA) medium. After an additional 3 d incubation in the dark at 18 °C, the cultures were sealed with Parafilm and placed upright under a black light (330–400 nm ultraviolet wave) for 7 d at 18 °C in the dark to induce sporulation. The spores were washed off with 2 mL of sterile water, and the suspension was filtered through three layers of lens tissue. The number of spores was counted using a hemocytometer under a microscope. Each treatment was conducted three times.

The resulting spore suspension was diluted with sterile water to 10^5^ spores/mL, and 200 µL of the diluted spore suspension was evenly spread on a PDA plate. After a 9 h incubation in the dark at 18 °C, spore germination was observed under a microscope. For each treatment, 100 spores were observed for germination. Each treatment was repeated twice, and the experiment was conducted twice.

### 2.8. Virulence of Procymidone-Resistant Mutants for Host Infection

Healthy grape fruits were selected with similar sizes and scratched to create a wound of about 3 mm in diameter at the same part of the grape skin of all fruits using a sterilized inoculation needle. A 5 mm mycelial plug of the strains was inoculated onto the wounded area. After culturing in the dark at 18 °C for 7 d, the lesion diameter was measured. Each treatment was conducted ten times, and the experiments were conducted twice.

### 2.9. Sensitivity of B. cinerea Strains to Osmotic Stress

The resistant mutants and their wild-type parent strains of *B. cinerea* were cultured in the dark at 18 °C for 3 d. Then, a 5 mm mycelial plug cut from the edge of the colonies was transferred to the PDA plates amended with 1 M glucose, 0.5 M NaCl, 1 M NaCl, and 1 M sorbitol, respectively. The colony diameter was measured after 3 d. Each treatment was conducted three times.

### 2.10. Sequence Analysis of the BcBos1 Gene and Other Target Genes between Procymidone-Resistant Mutants and Their Wild-Type Parental Strains

The DNA of *B. cinerea* was extracted using the sodium dodecyl sulfate (SDS) methods according to Watanabe’s methods [16], with minor modifications. The primers (Table 1) were designed according to the gene sequences and synthesized by Beijing Tsingke Biotech Co., Ltd. (Beijing, China) based on the DNA sequence of *BcBos1* (BCIN_01g06260), *SdhA*, *SdhB*, *SdhC*, *SdhD*, *Cytb*, and *Mrr1* of *B. cinerea* with the primers shown in Appendix A. 

The amplification reactions were performed in 25 µL of the reaction mixture containing 12.5 µL of 2 × Taq Master Mix, 1 µL of primer, 1 µL of template DNA, and we then added ddH_2_O to the 25 µL. The PCR products were examined by electrophoresis in a 1% agarose gel. Then, the following PCR parameters were used as follows: an initial preheating for 5 min at 95 °C, followed by 35 cycles of denaturation at 94 °C for 30 s, annealing at 60 °C for 30 s, an extension at 72 °C for 4.5 min, and a final extension at 72 °C for 10 min. The PCR was subsequently sequenced by the Beijing Tsingke Biotech Co., Ltd. (Beijing, China). SnapGene software 6.0.2 (GSL Biotech LLC, Chicago, IL, USA) was used to compare the amino acid sequences of the resistant mutants and sensitive strains.

### 2.11. Comparison of the Expression Level of BcBos1 between Procymidone-Resistant Mutants and Their Wild-Type Parental Strains

The expression of the *BcBos1* genes was assessed through a quantitative polymerase chain reaction (qPCR) conducted on an ABI7500 sequence detection system (Applied Biosystems, Waltham, MA, USA) using the FastSYBR Mixture Kit (Beijing ComWin Biotech Co., Ltd., Beijing, China) following Wu’s methods [17], with specific primers (Table 1) and cDNA as the templates. The program of the qPCR was set as denaturation at 95 °C for 2 min, followed by 40 cycles of 95 °C for 10 s and 60 °C for 34 s. The reaction system contained 10 µL of FastSYBR × 2, 0.4/0.4 µL of a mixture of forward/reverse primer, 1 µL of template cDNA, and 8.2 µL of ddH_2_O. The template cDNA was obtained with reverse transcription from RNA of B05.10 using a PrimeScript RT Reagent Kit with gDNA Eraser (Takara, Beijing, China). The relative expression of the genes was calculated using the 2^−∆∆Ct^ method. The actin gene (Bcin_16g02020) and *BcEF* (Bcin_09g05760) were used as a reference to normalize the quantification of the *BcBos1* expression levels. This experiment was conducted twice, and each experiment included three replicates for each treatment.

### 2.12. Molecular Docking of Procymidone at B. cinerea BcBos1 Site

The BcBosl and mutants of the three-dimensional model were predicted by AlphaFold through the online software ColabFold·v1.5.3. Procymidone was docked into the BcBos1 or mutants with the Autodock Tools 1.5.6 software. The relationship between the mutation site and the affinity of fungicides was analyzed on the basis of the energy score.

### 2.13. qRT-PCR of ABC Transporter Genes

Fourteen ABC transporter genes were measured in procymidone-resistant mutants and their parental strains. A qPCR assay was performed as described in Section 2.10, and the primers were shown in Table 1. The relative expression of the genes was calculated using the 2^−∆∆Ct^ method, and the actin gene was used as a reference to normalize the quantification of the ABC gene expression levels.

### 2.14. Statistical Analyses

Data were analyzed using GraphPad Prism 8.4.3 (GraphPad Software Inc., San Diego, CA, USA). Significance differences of the treatments were analyzed by one-way ANOVA; *p* values < 0.05 were considered to be significant.

## 3. Results

### 3.1. Procymidone-Resistant Mutants of B. cinerea

B05.10 and 223, the two parental isolates, were subjected to domestication by procymidone, but resistant mutants were only obtained from B05.10. The mutants from the 223 strain exhibit resistance to procymidone. However, this resistance gradually diminished when transferred to procymidone-free PDA plates after ten times. A total of nine procymidone-resistant mutants were highly resistant to procymidone, where the EC_50_ value changed from 0.13 to more than 100, and with resistant factors (RF) of >100.

Both the EC_50_ and RF of both the first and tenth generation of procymidone-resistant mutants were greater than 100, indicating all nine mutants were highly resistant to procymidone and that their resistance was stable.

### 3.2. Cross-Resistance of Procymidone-Resistant Mutants

Nine mutants were tested for resistance to fludioxonil, azoxystrobin, fluazinam, boscalid, difenoconazole, and pyrimethanil (Table 2). All mutants exhibited higher resistance to fludioxonil, with resistance factors ranging from 2 to 26, indicating a positive cross-resistance between fludioxonil and procymidone. Some mutants also showed high resistance to azoxystrobin, boscalid, difenoconazole, and pyrimethanil, with resistance factors ranging from 2.71 to 119.26, 1.65 to 6.21, 2.17 to 15.40, and 2.88 to 15.13, respectively. Particularly, the mutants, including Pro 2, 4, 6, 7, 9, and 10, developed resistances to more than two fungicides with a similar mode of action.

### 3.3. Fitness of B. cinerea Mutants

The mycelial growth of all mutants significantly slowed down after 72 h, showing a significant difference compared to the parental strain (Figure 1a). The sporulation ability of the resistant mutants was significantly reduced compared to the parent strain, with most of the resistant mutants not sporulating at all (Figure 1b). Spore germination was not conducted due to the lack of spores. Both the parental and the derived resistant strains were able to effectively infect grape fruits, causing typical symptoms of grape gray mold (Figure 1c). The virulence of all mutants significantly decreased.

The optimal growth temperature for *B. cinerea* was 25 °C. All mutant strains exhibited a slow growth at 4 °C, and Pro 10 showed nearly no growth at 30 °C. In general, the mutants had a reduced growth, although a few mutants exhibited a higher growth rate than the parent strain (Table 3).

Both mutants and their parental strains were sensitive to high salt osmotic pressure, but the mutants were more sensitive (Figure 2). The mycelial growth of B05.10 was inhibited on PDA containing 1M glucose, 1M sorbitol, and 0.5 M NaCl by 35.1%, 7%, and 37.6%, respectively. In contrast, all mutants exhibited significantly higher inhibition rates in response to these osmotic pressures, with the highest inhibition of 89.1%, 94.5%, and 99.3%.

### 3.4. Sequence Variation in the Target Genes

The *SdhA*, *SdhB*, *SdhC*, *SdhD*, *Cytb,* and *Mrr1* genes did not exhibit any mutations in the mutants. Mutants Pro 1 and Pro 2 only mutated at the 3605 bp position in the non-coding region. Mutants Pro 3, Pro 4, Pro 6, Pro 7, and Pro 8 possessed the mutation TGG→TGA, leading to a change from serine to a termination codon at position 647 (W647X). Mutant Pro 9 possessed CGA→TGA, leading to a change from arginine to a termination codon at position 96 (R96X). Mutant Pro 10 possessed the mutation CAG→TAG, leading to a change from glutamine to a termination codon at position 751 (Q751X) (Figure 3). The translation process of BcBos1 in these resistant mutants may be affected by these single-point mutations.

### 3.5. BcBos1 Gene Expression Using qRT-PCR

The expression of *BcBos1* in mutants and the parental strain was assessed by qPCR (Figure 4). Most mutants did not exhibit a significantly differential expression. Only Pro 3, Pro 6, and Pro 8 showed a slight upregulation, ranging from 2.09 to 6.61-fold compared to the parental strain.

### 3.6. Molecular Docking

In order to elucidate the resistance mechanism of the mutant, three BcBos1 proteins were predicted from AlphaFold for docking with procymidone. The molecular docking results revealed that procymidone and the sensitive-BcBos1 protein formed three simultaneous hydrogen bonds with LYS857, LEU922, and GLY-923, yielding a binding force score of −6.3 kcal/mol (Figure 5a). However, the binding force score between procymidone and the BcBos1 protein from resistant strains Pro 3 to 8, 9, and 10 were −6.3, −5.6, and −5.3 kcal/mol, respectively (Figure 5b–d). These findings suggest that the affinity between procymidone and BcBos1 diminished following serine, arginine, or glutamine changes to a termination codon.

### 3.7. Expression of ABC and MFS Transporter Genes Using qRT-PCR

The expression of fourteen ABC transporter genes in three mutants (Pro 2, Pro 6, and Pro 10) and their parental strain were assayed using the qPCR, with *Bcactin* and *BcEF* being reference genes (Figure 6 and Appendix A). *BcatrB*, *BcatrG*, *BcatrK*, and *BcmfsM2* were upregulated, ranging from 2 to 13.8-fold in mutants compared to the parental strain B05.10. Although *BcatrA*, *BcatrB*, *BcatrG*, *BcatrK*, and *mfsM2* were upregulated in B05.10, ranging from 2 to 93.7-fold after the addition of procymidone and azoxystrobin, the degree of upregulation in the mutants was significantly higher than that of their parents, ranging from 2 to 51-fold. Meanwhile, the expression levels of *BcatrA* and *BcatrG* showed a significant increase after azoxystrobin treatment compared to procymidone treatment. This indicates that the upregulation of ABC transporter genes, especially *BcatrB* and *BcatrG*, is involved in the resistance of the tested strains to procymidone even more in MDR.

## 4. Discussion

Multidrug resistance (MDR) of *B. cinerea* has notably merged in the field. We have generated nine laboratory mutants of *B. cinerea* that were simultaneously resistant to both procymidone and other fungicides having different MOAs. We have demonstrated that through procymidone domestication, *B. cinerea* can develop MDR, including the quinone-outside inhibitor (QoIs) fungicide azoxystrobin, 14α-demethylation (DMIs) fungicide prochloraz, succinate dehydrogenase inhibitor (SDHIs) fungicide boscalid, and anilinopyrimidine fungicide pyrimethanil, with resistance factors ranging from 1.65 to 119.26. This has not been reported in previous studies, but it is consistent with the MDR performance of the field-resistant strains reported that are resistant to fludioxonil and other fungicides [18,19,20,21,22]. Based on this observation, it is speculated that the use of procymidone in the field can be one possibility to induce MDR in *B. cinerea*.

Procymidone is frequently associated with resistant populations of *B. cinerea*. ABC transporters in *B. cinerea* are upregulated when exposed to fungicides such as fludioxonil and procymidone [13]. The two-component histidine kinase (HK) phosphorelay protein complexes, which were affected by procymidone, are key components for adapting to changing environmental conditions, regulating functions encompassing host recognition, virulence, stress, and hormonal response, as well as chemotaxis, quorum-sensing, and osmo-sensing. However, this occurs not only within the relationship between HK phosphorelay protein complexes and the regulation of ABC transporters, but it is also unclear whether procymidone directly triggers MDR.

The derived mutants exhibited resistance to both boscalid and azoxystrobin, showing the presence of MDR. *SdhA*, *SdhB*, *SdhC*, *SdhD*, and *Cytb* [5,23], as well as in the transcription factor *Mrr1* [13], are considered the sites for point-mutants corresponding to these two fungicides, but none of them were found to be mutated. ABC transporters have been reported to contribute to MDR. Specifically, *BcatrB*, *BcatrD*, and *BcatrK* are crucial for *B. cinerea* to efflux fungicides [2,9,13,24]. The ABC transporter genes *BcatrB*, *BcatrK*, and *BcatrG* were upregulated, especially after being treated in procymidone-resistant mutants. The overexpression of these efflux genes might lead to the efflux of fungicides, thereby reducing the amount of fungicides binding to the target and resulting in resistance.

In eukaryotes, intracellular signaling pathways rely on diverse signaling elements, such as mitogen-activated protein kinases (MAPKs). Among these, the Hog pathway holds significance for filamentous fungi, mediating responses to osmotic changes and influencing virulence [25]. In *B. cinerea*, several mutations in the two-component histidine kinase gene (*BcOS1*, *BcBos1*, or *Bos1*) responsible for DCF resistance have been identified [26]. The site mutations at codons 365 (I365N/S), 369 (Q369P), and 373 (N373S) were found to be linked to procymidone resistance in *B. cinerea* [27]. Reports associating resistance to procymidone with this pathway highlight mutations in the *BcBos1* gene. This research has identified three novel mutation sites (W647X, R96X, and Q751X) within the target gene. Furthermore, molecular docking analyses revealed that these mutations resulted in a reduced binding affinity to the target protein. This represents a newly discovered mechanism of target mutation.

Mitogen-activated protein kinase (MAPK) cascades play an important role in *B. cinerea*’s response to stress [28]. The high-osmolarity glycerol (HOG) pathway, a key MAPK pathway, contributes to oxidative and cell wall stress responses, with *BcBos1* initiating the HOG pathway [29]. The HOG pathway has been extensively studied in Saccharomyces cerevisiae. Hyperosmotic conditions lead to the dephosphorylation of a transmembrane HK (Sln1) that negatively regulates a MAPK cascade via the phosphorelay elements Ypd1/Ssk1. The MAPK cascade functions via the sequential phosphorylation of a Ssk2/Ssk22/Ste11 (MAPKKK)-Pbs2/Sty1 (MAPKK)-Hog1/Sak1 (MAPK), which migrates to the nucleus to modulate the activity of several transcription factors mediating the downstream transcriptional response [30], such as Atf1 [31] and Pap1 [32]. On the other hand, Skn7 phosphorylation most likely cooperates with the Yap1 transcription factor [33].

While this pathway is pivotal in conferring tolerance to phenylpyrrole and dicarboximide fungicides, its function in response to cell walls and other stresses might vary across different fungal species [34]. The unique histidine kinase Sln1, involved in the high-osmolarity response of filamentous ascomycetes, is termed Nik-1/Os-1 in *Neurospora crassa*, Nik1 in *Cochliobolus heterostrophus*, and Hik1 in *Magnaporthe grisea* [35]. In *B. cinerea*, *BcOs1* is necessary for resistance to the dicarboximide fungicide, conidiation, and virulence [25,36], and the BcOs5, BcSak1 (BcOs2), and BcRrg1 were also important for sensitivity to these fungicides [37,38,39]. The transcription factors FgCrz1, which regulated multidrug transporters, membrane lipid biosynthesis, and metabolism, were influenced by the calcium-calcineurin and HOG pathways [40]. Thus, the signaling pathway was predicted to play a significant regulatory role in the expression of ABC transporters. Mutations of *BcBos1* genes in this pathway may be the main reason for the ABC transporters’ upregulation.

The target gene *BcBos1,* associated with procymidone resistance, is linked to the HOG-MAPK pathway and ABC transporters, forming a potential mechanism of resistance to procymidone in the obtained mutants. However, the precise relationship between site mutation and ABC transporter genes in *B. cinerea* still remains to be fully elucidated. Moreover, it remains to be determined whether the fungicide procymidone or alterations to its target protein BcBos1, as well as modifications in the HOG-MAPK signaling pathway, may lead to resistance or even MDR in *B. cinerea*.

## 5. Conclusions

With the domestication of procymidone, *Botrytis cinerea* produced highly resistant mutants to the osmotic signal inhibitors procymidone and fludioxonil and also developed MDR to fungicides with different modes of action, such as QoI, DMI, uncoupler, SDHI, and anilinopyrimidine. The mutations in target gene *BcBos1* and the upregulation of ABC transporter coding genes form a potential mechanism of the two kinds of resistances in the obtained mutants. Moreover, it remains to be determined how the protein BcBos1 in the HOG-MAPK signaling pathway is linked to MDR in *B. cinerea*.

## Figures and Tables

**Figure 1 jof-10-00261-f001:**
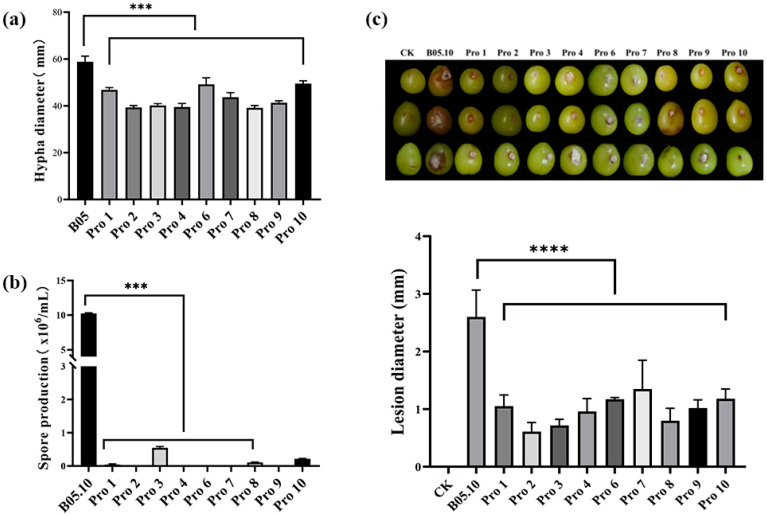
Fitness of procymidone-resistant mutants and parental strain of *Botrytis cinerea*, including (**a**) mycelial growth rate, (**b**) spore production, and (**c**) virulence (‘***’ indicates *p* value < 0.001; ‘****’ indicates *p* value < 0.0001).

**Figure 2 jof-10-00261-f002:**
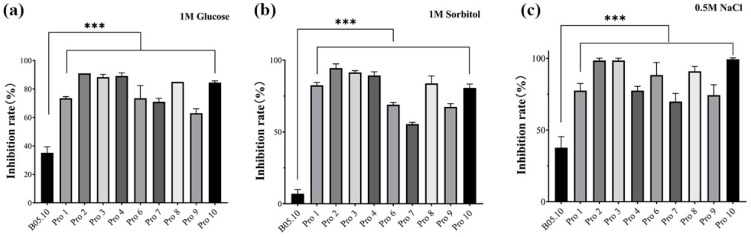
Sensitivity of procymidone-resistant mutants and parental strain of *Botrytis cinerea* to osmotic stress, including (**a**) 1 M glucose, (**b**) 1 M sorbitol, and (**c**) 0.5 M NaCl (‘***’ indicates *p* value < 0.001).

**Figure 3 jof-10-00261-f003:**
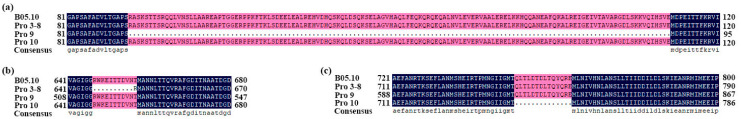
*BcBos1* sequencing of procymidone-resistant mutants and parental strain of *Botrytis cinerea*. (**a**–**c**) Alignment of the partial deduced amino acid sequence of BcBos1 from B05.10 strains and procymidone-resistant mutants. Exonic regions of genes are indicated with blue rectangle, and introns are indicated with gray line in (**a**–**c**).

**Figure 4 jof-10-00261-f004:**
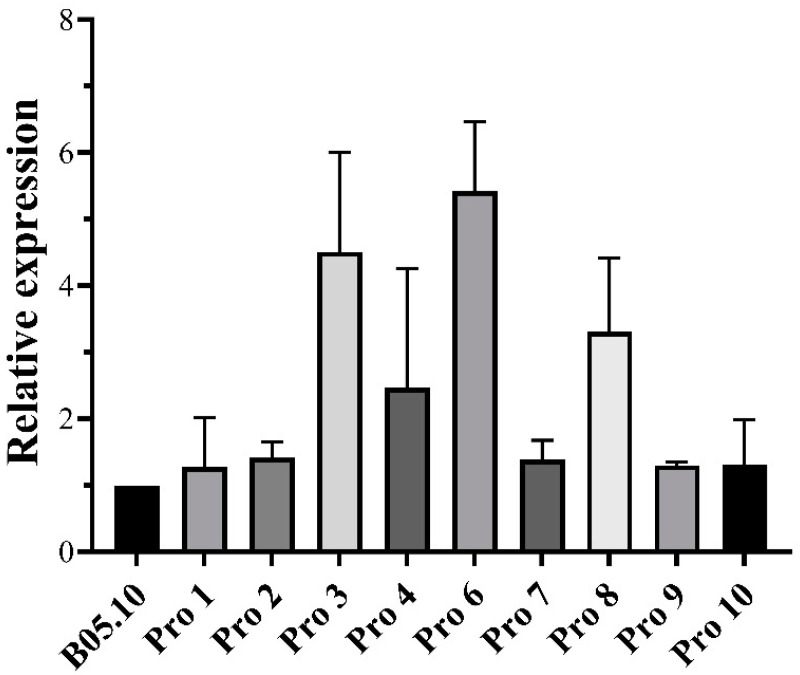
Quantitative polymerase chain reaction (qPCR) analysis on *BcBos1* genes in procymidone-resistant mutants of *Botrytis cinerea*.

**Figure 5 jof-10-00261-f005:**
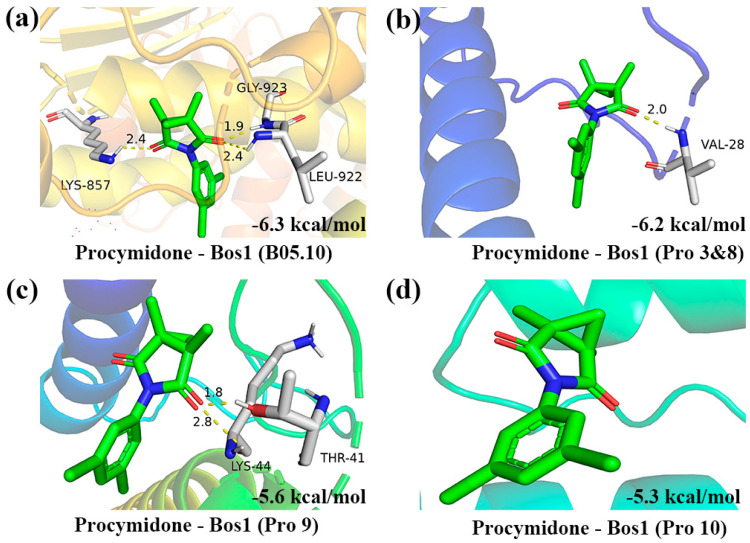
Molecular docking of procymidone to BcBos1 proteins from procymidone-resistant mutants and their sensitive parental isolates in *Botrytis cinerea*. (**a**) The binding between procymidone and wild-type BcBos1; (**b**) the binding between procymidone and BcBos1 of Pro 3 &8; (**c**) the binding between procymidone and BcBos1 of Pro 9; and (**d**) the binding between procymidone and BcBos1 of Pro 10.

**Figure 6 jof-10-00261-f006:**
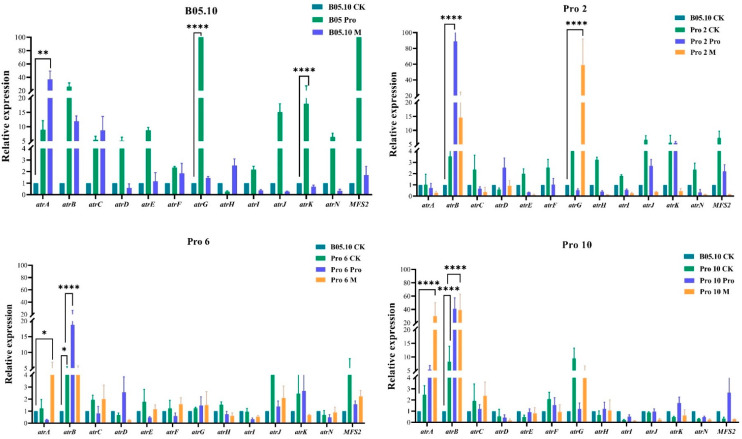
Quantitative polymerase chain reaction (qPCR) analysis of 14 ABC transporter genes in procymidone-resistant mutants with *Bcactin* being a reference gene. CK: non-treated; Pro: procymidone added; M: azoxystrobin added; ‘****’ indicates *p* value < 0.0001; ‘**’ indicates *p* value < 0.001; ‘*’ indicates *p* value < 0.05.

**Table 1 jof-10-00261-t001:** Primers used for quantitative polymerase chain reaction (qPCR) assay.

Genes	Sequences
*BcActin*	F: TCCAAGCGTGGTATTCTTACCC
	R: TGGTGCTACACGAAGTTCGTTG
*BcEF*	F: TCCTTCAAGTACGCATGGGT
	R: GTACCAGCGGCAATGATGAG
*BcatrA*	F: CTGGACGAGGCTACTTCTGG
	R: TGTACTCCACGGTCACCAAA
*BcatrB*	R: AGAGAGGGGTTGCGAATTCA
	R: AGAGAGGGGTTGCGAATTCA
*BcatrC*	F: TTTGGAATCCAGAAGCAACC
	R: TTCTTCGTGGCCTTTGTTCT
*BcatrD*	F: TCCAGGAGCCAGCAATACAA
	R: AACCCTGCGGCAAATGAATT
*BcatrE*	F: ACAATCATCTGCGGGAAAAC
	R: GAATCTGTGCAACGAAAGCA
*BcatrF*	F: AGGGGCAAGACACTTTTGTG
	R: GCCCTTGCAAAATCCTCTTT
*BcatrG*	F: AGATACTCGGCGTTGCTTGT
	R: TTTGGCAAAAAGGACGAAAG
*BcatrH*	F: GGACAACGTGCAAAGATTCA
	R: TGCCTGTTCCGTGTGTGTAT
*BcatrI*	F: TTAGATGCCGAATCCGAAAC
	R: AATTTCGAAAAGGCCGAGTT
*BcatrJ*	F: CGCTTATCAAGAGCACACCA
	R: GCCATTCAAATGTGGGAATC
*BcatrK*	F: CCGCTTTGATGGAGAACGAG
	R: GTGATGTAGTCGCCACCAAC
*BcatrN*	F: ACTCCATCCCCAATCGAAA
	R: AGTGATTGGGCAACTGACA
*Bcmfs2*	F: CATGGCTGTCTCATTCGGTG
	R: GGAATGAAGATGGCGGTTCC
*BcBos1*	F: GATGTGGGTGTGGATGGTAAGATGG
	R: TGCATCATGATTCTCATTTATTCTCAT

F: forward primer. R: reverse primer.

**Table 2 jof-10-00261-t002:** Cross-resistance assay of *Botrytis cinerea* mutants (Pro 1 to Pro 10) and their wild-type parental strain by measuring the effective concentration for 50% growth inhibition of mycelia [EC5_0_ (μg/mL)] to test fungicides.

Strain	EC_50_ (μg/mL)						
	Procymidone	Fludioxonil	Azoxystrobin	Fluazinam	Boscalid	Difenoconazole	Pyrimethanil
B05.10	0.13	0.03	0.31	0.01	1.17	1.41	0.08
Pro 1	>100	>100	1.10	0.08	0.25	0.91	0.24
Pro 2	>100	>100	1.79	0.16	0.30	10.54	1.07
Pro 3	>100	>100	0.84	0.10	0.67	3.06	0.23
Pro 4	>100	>100	23.02	0.26	5.03	21.71	0.43
Pro 6	>100	>100	36.97	0.23	6.30	4.94	0.32
Pro 7	>100	>100	3.79	0.11	4.99	1.31	0.74
Pro 8	>100	>100	<0.01	0.18	7.26	0.23	0.16
Pro 9	>100	>100	0.21	0.10	1.93	4.31	1.21
Pro 10	>100	>100	7.30	0.02	6.27	0.71	0.83

**Table 3 jof-10-00261-t003:** Mycelial growth of procymidone-resistant mutants and parental strain of *Botrytis cinerea* at various temperatures.

Isolate	Mycelial Growth (mm)				
	4 °C	14 °C	18 °C	25 °C	30 °C
B05.10	17.00 ± 1.73 d *	39.00 ± 1.00 cde	42.50 ± 3.27 cd	58.50 ± 6.68 ab	43.00 ± 4.12 a
Pro 1	25.00 ± 3.00 abc	40.00 ± 3.74 cde	40.75 ± 3.83 cd	46.50 ± 1.65 abc	41.50 ± 4.55 a
Pro 2	28.00 ± 2.00 a	48.50 ± 0.86 b	52.75 ± 0.82 ab	49.00 ± 1.00 abc	21.00 ± 1.00 d
Pro 3	21.50 ± 1.65 bcd	45.50 ± 2.59 bc	43.75 ± 0.43 bc	56.50 ± 6.53 ab	39.50 ± 0.86 ab
Pro 4	24.00 ± 1.41 abc	45.00 ± 3.60 bcd	44.50 ± 5.12 bc	50.50 ± 2.17 abc	29.00 ± 1.73 c
Pro 6	26.50 ± 1.65 ab	47.00 ± 2.23 b	47.50 ± 2.59 abc	56.00 ± 6.48 ab	33.00 ± 3.31 bc
Pro 7	23.00 ± 1.00 abc	39.00 ± 1.00 cde	42.00 ± 1.41 cd	47.00 ± 1.73 abc	34.00 ± 1.41 bc
Pro 8	20.50 ± 2.59 cd	38.50 ± 2.59 de	39.00 ± 4.12 cd	44.50 ± 10.52 bc	28.00 ± 2.00 cd
Pro 9	26.50 ± 0.86 ab	34.00 ± 1.41 e	33.50 ± 1.65 d	39.00 ± 5.19 c	30.00 ± 2.00 c
Pro 10	22.00 ± 1.41 bcd	57.50 ± 2.59 a	55.00 ± 6.08 a	59.50 ± 0.86 a	6.75 ± 0.43 e

* Values followed by the different letters within a column represent significant difference between mutants and their corresponding parental isolates (*p* < 0.05).

## Data Availability

Data are contained within the article.

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
