# Peer review of "Procymidone Application Contributes to Multidrug Resistance of Botrytis cinerea"

_jof, 2024, doi:10.3390/jof10040261_

Round 1
Reviewer 1 Report
The manuscript titled "Contribution of Procymidone Application to Multidrug Resistance in Botrytis cinerea" presents a thorough comprehension of resistant mutations in Botrytis cinerea related to procymidone. It’s a very complex comprehensive study explaining the fungicide resistance characteristics related to fungicide procymidone.
|
|
|
Some points that should be clarify/added:
- In the Materials and Methods section, the authors note the utilization of Botrytis cinerea strains B05.10 and 223, designating B05.10 as the standard strain and 223 as the sensitive strain. They should detail the methodology utilized to determine the sensitivity of these strains, as well as provide information regarding the origin of these strains.
- Additionally, it’s for recommend that the authors include a Conclusion section in their manuscript to highlight results obtained in the study, as well as further perspectives.
Author Response
Reviewer #1:
The manuscript titled "Contribution of Procymidone Application to Multidrug Resistance in Botrytis cinerea" presents a thorough comprehension of resistant mutations in Botrytis cinerea related to procymidone. It’s a very complex comprehensive study explaining the fungicide resistance characteristics related to fungicide procymidone.
Some points that should be clarify/added:
- In the Materials and Methods section, the authors note the utilization of Botrytis cinerea strains B05.10 and 223, designating B05.10 as the standard strain and 223 as the sensitive strain. They should detail the methodology utilized to determine the sensitivity of these strains, as well as provide information regarding the origin of these strains.
Response: The sentence was modified as at Line 85-87:
“Botrytis cinerea strains B05.10 (obtained from Germany, Institut für Botanik, Westfälische Wilhelms-Universität, Münster) and 223 (collected from Yinlong Soil Farm in Shanghai in April 2016) were used as a standard resistant and sensitive strain, respectively.”
- Additionally, it’s for recommend that the authors include a Conclusion section in their manuscript to highlight results obtained in the study, as well as further perspectives.
Response: The Conclusion section was added as follow:
“The target gene BcBos1, associated with procymidone resistance, is linked to the HOG-MAPK pathway and ABC transporters, forming a potential mechanism of resistance to procymidone in the obtained mutants. However, the precise relationship between site mutation and ABC transporter genes in B. cinerea still remains to be fully elucidated. Moreover, it remains to be determined whether the fungicide procymidone or alterations to its target protein BcBos1, and modifications in the HOG-MAPK signaling pathway, may lead to resistance or even MDR in B. cinerea.”
Reviewer 2 Report
The authors report the involvement of Procymidone in the process of multi-drug resistance development in Botrytis cinerea. This is an interesting well-planned and properly structured study, with only a few minor issues requiring correction. I would like to mention that the usage of single genes as a reference for qPCR is currently considered as insufficient as many housekeeping genes are not as stably expressed as it has been previously considered. Therefore in that case I would suggest caution in the interpretation of such data if the overexpression is not extremely high. Please, however, bear that in mind in your future studies. Please carefully check the names of genes and proteins and write the names accordingly. I consider that the presented manuscript deserves to be published in the Journal of Fungi after minor revision.
Line 14: Please rephrase this sentence as it seems contradictive to the next one
Line 20: induction of multidrug resistance
Line 21: DNA sequencing cannot confirm overexpression
Line 23: Please follow the rules of genes and protein name-writing
Line 25 It is unclear what B05.10 is yet
Line 63 It is unclear not skeptical
Line 87: If they are both sensitive why one is sensitive and the other is resistant, please explicitly explain why this strain was selected.
Line 90: Please explain how the transfer to increasing procymidone concentration was carried out, how many transfers, and what with the different starting concentrations all the plugs were transferred to 200 or each time the concentration was doubled.
Line 120: Please keep one system ℃ line 84 has ℃ please unify the fonts
Line 127: Please provide the irradiation parameters
Line 154: provide primer sequence in the supplementary information
Line 187: Please confirm the ANOVA assumptions before using this test
Line 296: Unclear sentence
Line 335: Molecular docking is an insufficient method to support this statement fully please rephrase.
Author Response
Reviewer #2:
The authors report the involvement of Procymidone in the process of multi-drug resistance development in Botrytis cinerea. This is an interesting well-planned and properly structured study, with only a few minor issues requiring correction. I would like to mention that the usage of single genes as a reference for qPCR is currently considered as insufficient as many housekeeping genes are not as stably expressed as it has been previously considered. Therefore, in that case I would suggest caution in the interpretation of such data if the overexpression is not extremely high. Please, however, bear that in mind in your future studies. Please carefully check the names of genes and proteins and write the names accordingly. I consider that the presented manuscript deserves to be published in the Journal of Fungi after minor revision.
Response: The qPCR results of BcEF used as reference gene were added in Supplement Fig.1.
Line 14: Please rephrase this sentence as it seems contradictive to the next one
Response: The sentence at Line 13-15 was modified as
“Although the resistance to procymidone has been observed in the field, it remains uncertain whether procymidone is a multidrug resistance (MDR)-inducing factor.”
Line 20: induction of multidrug resistance
Response: The sentence was modified as
“indicating the induction of MDR”
Line 21: DNA sequencing cannot confirm overexpression
Response: The sentence was modified as
“Transcriptive expression analysis using quantitative polymerase chain reaction (qPCR) revealed that the mutants overexpressed ABC transporter genes, ranging from 2 to 93.7-fold. These mutants carried single-point mutations W647X, R96X and Q751X within BcBos1 by DNA sequencing.”
Line 23: Please follow the rules of genes and protein name-writing
Response: All names of genes and protein were revised.
Line 25 It is unclear what B05.10 is yet
Response: It has been revised at Line 25 and Line 85.
Line 63 It is unclear not skeptical
Response: It has been revised.
Line 87: If they are both sensitive why one is sensitive and the other is resistant, please explicitly explain why this strain was selected.
Response: Both the B05.10 and 223 strains were screened using procymidone, but only the B05.10 strain yielded stable mutants. The mutants from 223 strain exhibit resistance to procymidone. But this resistance diminished gradually when transferred to procymidone-free PDA plates after ten times. This observation has been added in the manuscript at Line 203-205.
Line 90: Please explain how the transfer to increasing procymidone concentration was carried out, how many transfers, and what with the different starting concentrations all the plugs were transferred to 200 or each time the concentration was doubled.
Response: It has been revised: “In each transfer, the procymidone concentration was gradually increased from 200, 400 to 600 μg/mL and each concentration was substituted once.”
Line 120: Please keep one system ℃ line 84 has ℃ please unify the fonts
Response: It has been revised.
Line 127: Please provide the irradiation parameters
Response: It has been revised: “…black light (330-400nm ultraviolet wave).”
Line 154: provide primer sequence in the supplementary information
Response: It has been added in Supplementary table 1.
Line 187: Please confirm the ANOVA assumptions before using this test
Response: It has been revised:
“Significance differences of treatments were analyzed by the one-way ANOVA, P values < 0.05 were considered to be significant.”
Line 296: Unclear sentence
Response: The sentence was modified: “We have generated nine laboratory mutants of B. cinerea that were simultaneously resistant to both procymidone and other fungicides having different MOAs.”
Line 335: Molecular docking is an insufficient method to support this statement fully please rephrase.
Response: The sentence was modified: “Furthermore, molecular docking analyses revealed that these mutations result in a reduced binding affinity to the target protein. This represents a newly discovered mechanism of target mutation.”